# Possibly Invasive New Bioenergy Crop *Silphium perfoliatum*: Growth and Reproduction Are Promoted in Moist Soil

**L. Marie Ende *** , **Katja Knöllinger, Moritz Keil, Angelika J. Fiedler and Marianne Lauerer**

Ecological Botanical Gardens, Bayreuth Center for Ecology and Environmental Research (BayCEER), University of Bayreuth, 95447 Bayreuth, Germany; katja.knoellinger@uni-bayreuth.de (K.K.); moritz.keil@uni-bayreuth.de (M.K.); angelika.j.fiedler@uni-bayreuth.de (A.J.F.); marianne.lauerer@uni-bayreuth.de (M.L.)
* Correspondence: marie.ende@uni-bayreuth.de

**Abstract:** The cup plant (*Silphium perfoliatum*) is a new and promising bioenergy crop in Central Europe. Native to North America, its cultivation in Europe has increased in recent years. Cup plant is said to be highly productive, reproductive, and strongly competitive, which could encourage invasiveness. Spontaneous spread has already been documented. Knowledge about habitat requirements is low but necessary, in order to predict sites where it could spontaneously colonize. The present experimental study investigates the growth and reproductive potential of cup plant depending on soil moisture, given as water table distance (WTD). In moist soil conditions, the growth and reproductive potential of cup plant were the highest, with about 3 m plant height, 1.5 kg dry biomass, and about 350 capitula per plant in the second growing season. These parameters decreased significantly in wetter, and especially in drier conditions. The number of shoots per plant and number of fruits per capitulum were independent of WTD. In conclusion, valuable moist ecosystems could be at risk for becoming invaded by cup plant. Hence, fields for cultivating cup plant should be carefully chosen, and distances to such ecosystems should be held. Spontaneous colonization by cup plant must be strictly monitored in order to be able to combat this species where necessary.

**Keywords:** bioenergy crop; cup plant; groundwater; growth; invasive potential; reproductive potential; *Silphium perfoliatum*; soil moisture; water table distance

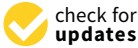



## 1. Introduction

In Europe, biogas is being increasingly produced as a renewable energy source to replace fossil fuels [1]. Currently maize (*Zea mays* L.) is the most dominant biogas crop, though its cultivation goes along with great ecological damage from the high application of machines, fertilizer, and pesticides. Therefore, alternative bioenergy crops are being sought that are more ecologically agreeable [2–5].

One promising alternative crop in this context is the cup plant (*Silphium perfoliatum* L.) [2]. This perennial, yellow-flowering C3-plant belongs to the Asteraceae family. It develops stems and flowers from the second year onwards and persists many years [6]. Native in the prairies of eastern North America, cup plant was introduced to Europe in the 18th century as an ornamental plant [6]. Since 2004 it has been used as a bioenergy crop in Germany [2], and as of 2019 about 4500 ha have been cultivated there [7]. Many other European countries are cultivating this crop for bioenergy as well [3].

Cup plant has many ecological advantages over maize [2]: It can be harvested profitably for more than 15 years [8], and the application of machines and pesticides is much lower compared to maize, an annual plant [2]. In soil, higher portions of microbial biomass, higher microbial diversity, and higher biological activity comparative to maize have been proven [4]. Benefits for many pollinator species have also been detected: Insects are strongly attracted to the flowers of cup plant, which have a long flowering period relatively late in the year when most other floral resources have already finished blooming [9–11].

Furthermore, cup plant is easy to cultivate, highly productive with a high biogas yield, and it is competitive and very reproductive [2,6]. These traits make it an attractive bioenergy crop. However, in combination with the fact that harvest in agriculture usually takes place after flowering, they carry the risk of spontaneous spreading and settlement out from the fields. Spontaneous colonization was already documented in Upper Franconia (Germany) [12] and other parts of Germany as well as in other European countries (e.g., Austria, Switzerland, Poland) [13–16]. In the Netherlands and Russia, cup plant has already been graded as "potentially invasive" [17,18]. Studies on the invasive potential of this species are essential and should be of interest for all involved stakeholders before cup plant is cultivated on a large scale.

For this purpose, comprehensive knowledge about site requirements is necessary to allow predictions about where cup plant might establish itself. However, little is known about its site preferences in Europe, especially regarding soil moisture. It is assumed that cup plant prefers soils with good moisture provision but is also fairly drought tolerant [3,5,19,20]. So far, there are mainly empirical data or assumptions and only few experimental studies about the yield of cup plant in Central Europe depending on soil moisture. In its native range in North America, cup plant colonizes moist bottomlands and floodplains near streambeds [6]. Assuming that cup plant grows and reproduces in Central Europe in the same way that it does in its native range, it carries a special risk of invasion on moist sites. It is known that these are often ecosystems of high value for nature conservation in Central Europe. To assess the risk of these ecosystems becoming colonized by cup plant, we executed a growth experiment with cup plant over two years in tanks similar to those of Ellenberg's Hohenheimer groundwater table experiment [21–23] at the Ecological Botanical Gardens of the University of Bayreuth, Germany. The question was: How do growth and reproductive potential of cup plant differ depending on groundwater level? This study will not only provide insights into the demands and autecology of cup plant for the first time; the approach is also innovative because the findings are of great interest for nature conservation as well for agriculture.

## 2. Materials and Methods

### 2.1. Experimental Setup

The experiment was carried out from May 2018 to September 2019 in tanks similar to those of Ellenberg's Hohenheimer groundwater table experiment [21–23] at the Ecological Botanical Gardens of the University of Bayreuth (Germany, Bavaria). Temperature in the first growing season (May to August 2018) ranged from 1 °C to 35 °C (mean 18 °C) and precipitation sum was 151 mm, and in the second growing season (May to August 2019) between −3 °C and 37 °C (mean 17 °C) and 195 mm, respectively. Seeds of cup plant (Metzler & Brodmann Saaten GmbH, Ostrach, Germany, harvested 2016, pretreated) were sown on 5 March 2018. Seedlings were pricked out three weeks later and cultivated in a greenhouse. On 7 May 2018 the experiment started by planting the saplings into four tanks. For pricking and planting we chose vital plants of equal and mean size.

Each of the four tanks was a south-exposed, 6.4° inclined concrete tank (8 m × 4 m), with a constant soil depth of 90 cm (Figure 1). Substrate was a homogeneous mixture of 40% native soil, 40% compost and 20% quartz sand. In the lower part of each tank water was supplied via a garden hose and a perforated plastic pipe. Excess water could drain through a hole in the tank wall (Figures 1 and 2). The water table was held constant by hand in the first season and automatically by a float switch in the second growing season. Therefore, the plants in the tanks had different water table distances (WTD). In each tank, plants were arranged in nine rows indicating different WTD and in each row, there were nine plants. Distance between rows was 90 cm and between plants in a row 30 cm. For data collection we excluded all margin plants, resulting in seven rows of 28 plants each, divided across the four tanks. After the first growing season, we harvested in each row and in each tank the aboveground biomass of the second, the fourth and the sixth plant (seen from west), resulting in $n = 12$ per treatment (= row) (Figure 2). Afterwards, we removed

the central part of the rootstock of these three and of the eighth plant. Consequently, in the second growing season five plants per row were left with distances of 60 cm between the plants. Excluding the margin plants, we had *n* = 12 per treatment (=row) again.

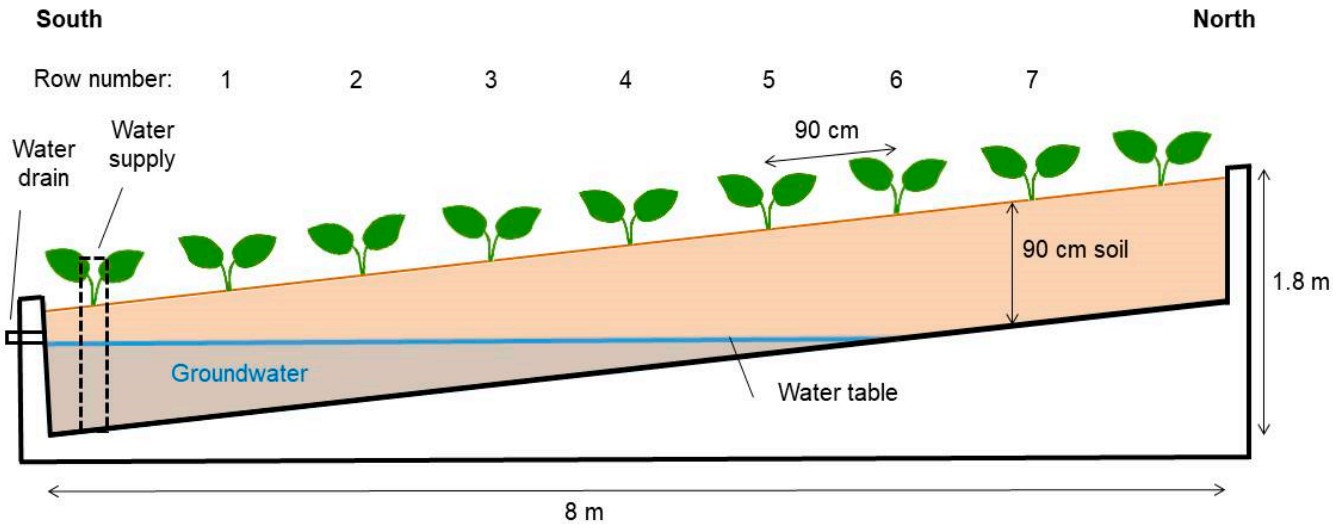

**Figure 1.** Scheme of a groundwater tank in longitudinal section.

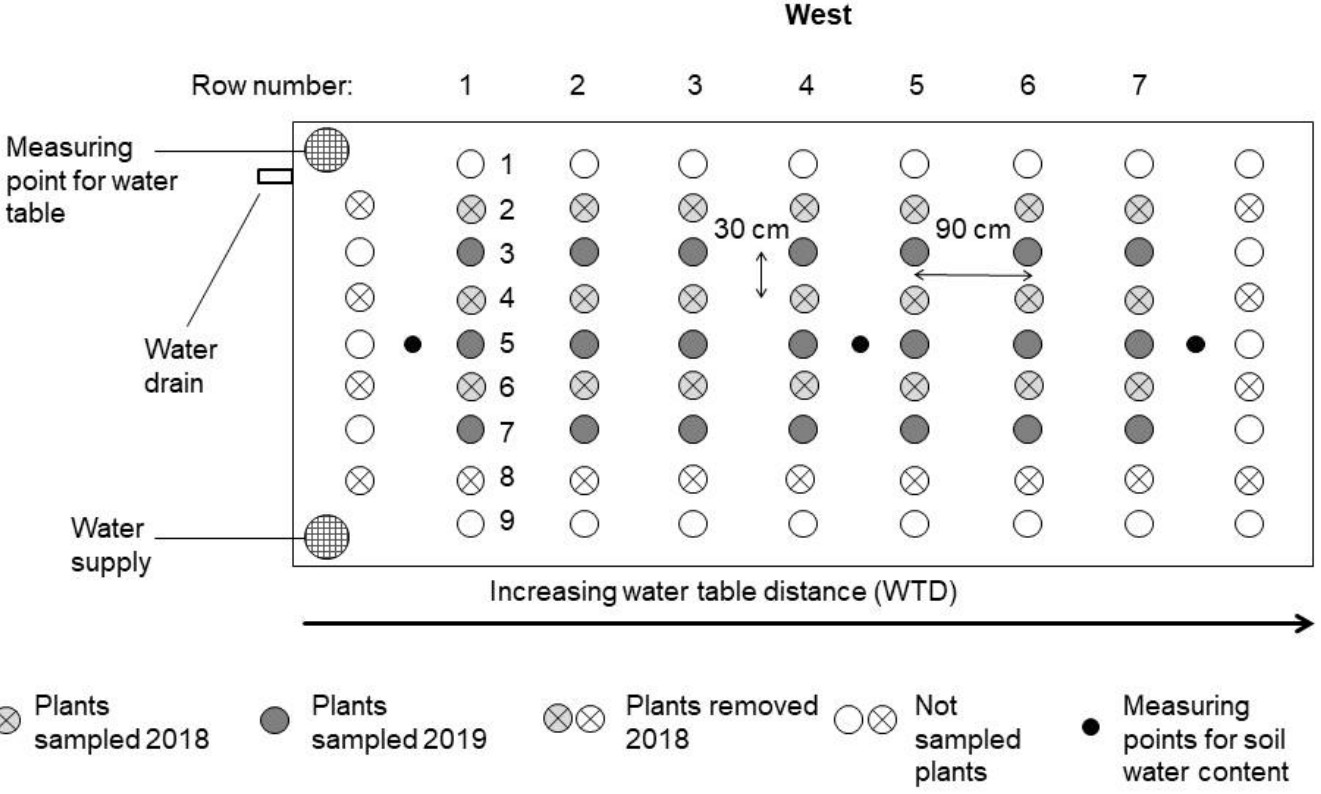

**Figure 2.** Top view of a groundwater tank indicating the plant arrangement, the harvesting scheme, and the measuring points for soil water content and water table.

### 2.2. Data Collection

In the first year, we harvested the living aboveground biomass on 9 August 2018, dried it in an oven at 70 °C until weight was constant, and measured the biomass with scales (PM 4600 Delta Range, Mettler-Toledo GmbH, Greifensee, Switzerland, same scales

for all further weight measurement unless otherwise noted). Two samples of two different treatments had to be discarded because of plant material loss; consequently, the sample number for biomass of the first growing season was 82 instead of 84. In November 2018, the number of shoots higher than 15 cm of the individuals left in the tanks was counted.

In the second year, sampling and harvesting were carried out between 10 and 13 September 2019. We measured plant height by calculating the mean of the five highest shoots. The number of shoots higher than 50 cm was counted for each plant and was assigned to one of the three stages (1) dead, when more than 50% of biomass was brown, (2) vegetative, without capitula or with buds less than 1 cm in diameter, and (3) generative, with buds of at least 1 cm diameter or flowering or fruiting capitula, respectively. We harvested a representative subsample of the shoots, noting the shoot stages of the subsample, which included at a minimum one-third of the shoots from the original sample. Rosette leaves and shoots lower than 50 cm were harvested completely. The subsamples of the shoots higher than 50 cm were split in compartments of dead and living biomass whose fresh weights were measured. If necessary, again a subsample was taken and its fresh weight was measured before the biomass was dried in an oven at 90 °C until constant weight. Dry weight of biomass was measured and extrapolated to total biomass per plant. Before drying, capitula of the subsets were counted, assigned to the three phenological stages (1) budding, from a diameter of 1 cm on, (2) flowering, when ray florets were visible, and (3) fruiting, when ray florets were fallen off (comprising beginning of fruit development until fallen-off fruits) and extrapolated to the whole plant.

Additionally, we harvested three ripe capitula of the remaining shoots of each plant and dried them separately in paper bags at room temperature. We counted their number of fruits and weighed them with scales (AE 240, Mettler-Toledo GmbH, Greifensee, Switzerland).

Since 12 July 2018, water table was automatically recorded in a perforated pipe in the lower part of each of the four tanks (Figure 2) every 10 minutes by a pressure sensor (BayEOS HX711 Board, BayCEER, Bayreuth, Germany). For data analyses, we averaged water table for the time from the beginning of the water table measurement to particular data sampling for each tank. WTD was calculated using the mean water table, the inclination of the tanks (6.4°), and the distances to water table sensor for each row of each tank. Because soil depth was only 90 cm, plants with a calculated WTD larger than 90 cm had no vertical water access.

Soil water content was measured weekly in the second growing season (May to September 2019) with a TDR probe (TRIME®-FM3, IMKO Micromodultechnik GmbH, Ettlingen, Germany) in plastic pipes on three positions (Figure 2) and two depths (5–25 cm and 40–60 cm) in each tank. In November 2019, we took soil samples in 30 cm soil depth on two positions of each tank to assess the relationship between soil water content and soil water tension. We took an undisturbed soil sample with a core cutter of 100 $cm^3$ and a disturbed soil sample of about 200 $cm^3$. The undisturbed soil samples were saturated with water over five days and afterwards dried in an oven at 105 °C until constant weight. The disturbed soil samples were filled each in two sampling rings of 20 $cm^3$, placed in a pressure pot for 26 days at −15,000 hPa (pF = 4.2), and dried until constant weight. Before and between these steps, all soil samples were weighed each time (PB 3002 DeltaRange, Mettler-Toledo GmbH, Greifensee, Switzerland). After these steps, soil samples were weighed again (PG 503-S DeltaRange, Mettler-Toledo GmbH, Greifensee, Switzerland).

Meteorological data were obtained by a weather station in the Ecological Botanical Gardens 310 m away from the experimental site operated by the Micrometeorology group, Prof. Dr. Thomas, BayCEER, University of Bayreuth.

The data on which calculations in this study are based are available in the supplementary materials (Tables S1–S7).

*2.3. Statistics*

Data analysis and plot presentation were executed with R version 3.6.1 [24]. Calculating means of data by treatment, we used the function "ddply" from the R package "plyr" version 1.8.4 [25]. To read climate data and groundwater level data we used the R package "bayeos" version 1.4.6 [26]. We used linear models (LM) and checked the diagnostic plots. In case of non-normal distribution or heteroscedasticity of residuals we tried generalized linear models (GLM). If both LM and GLM were not possible, we executed Spearman's rank correlation analysis or the Kruskal–Wallis rank sum test (Kruskal test) with the post-hoc test multiple comparison test after Kruskal–Wallis (KruskalMC) of the R package "pgirmess" version 1.6.9 [27]. The four tanks were considered as four blocks in a block design. We checked the influence of block (tank) with an LM respectively a GLM. In case of non-significance we eliminated the block for the final model. In case of a significant effect of block we exerted a mixed effect model with block as random factor using the R package "lme4" version 1.1–21 [28]. Fits of mixed effect models were built using the mean of intercepts. Level of significance was always 0.05.

**3. Results**

*3.1. Soil Water Conditions*

The treatments of the experiment created by the rows in the tanks with increasing water table distance (WTD) described a wide range of soil water conditions (Table 1). Because soil depth was only 90 cm the two driest rows (6 and 7, Figure 1) had no direct vertical access to water table. Logically, soil water content decreased with increasing WTD, as well near soil surface as in deep soil layer. Water content of waterlogged soil was $50 \pm 2\%$ vol (mean $\pm$ standard deviation). Permanent wilting point (pF-value = 4.2) was reached at $9 \pm 2\%$vol water content.

**Table 1.** Soil water conditions depending on the treatments. Row number in tank is counted from the bottom up (Figure 2). Water table distance (WTD) is given as mean $\pm$ standard deviation for both years separately. Soil depth was 90 cm. Soil water content was measured weekly only in the second growing season (year 2019) at three positions in the tanks (see Figure 2). Given values for each row were calculated by the models described in Figure 3.

| Row Number in Tank | First Year (2018) | Second Year (2019) | | Classification |
| --- | --- | --- | --- | --- |
| | WTD (cm) | WTD (cm) | Soil Water Content (%vol) in Depths | |
| | | | 5–25 cm | 40–60 cm | |
| 1 | $41 \pm 7$ | $40 \pm 11$ | 38 | 57 | wet |
| 2 | $51 \pm 7$ | $50 \pm 11$ | 31 | 48 | very moist |
| 3 | $61 \pm 7$ | $60 \pm 11$ | 26 | 40 | slightly moist |
| 4 | $71 \pm 7$ | $70 \pm 11$ | 22 | 33 | fresh |
| 5 | $81 \pm 7$ | $80 \pm 11$ | 18 | 28 | slightly dry |
| 6 | $91 \pm 7$ | $90 \pm 11$ | 15 | 23 | medium dry |
| 7 | $101 \pm 7$ | $101 \pm 11$ | 12 | 19 | rather dry |

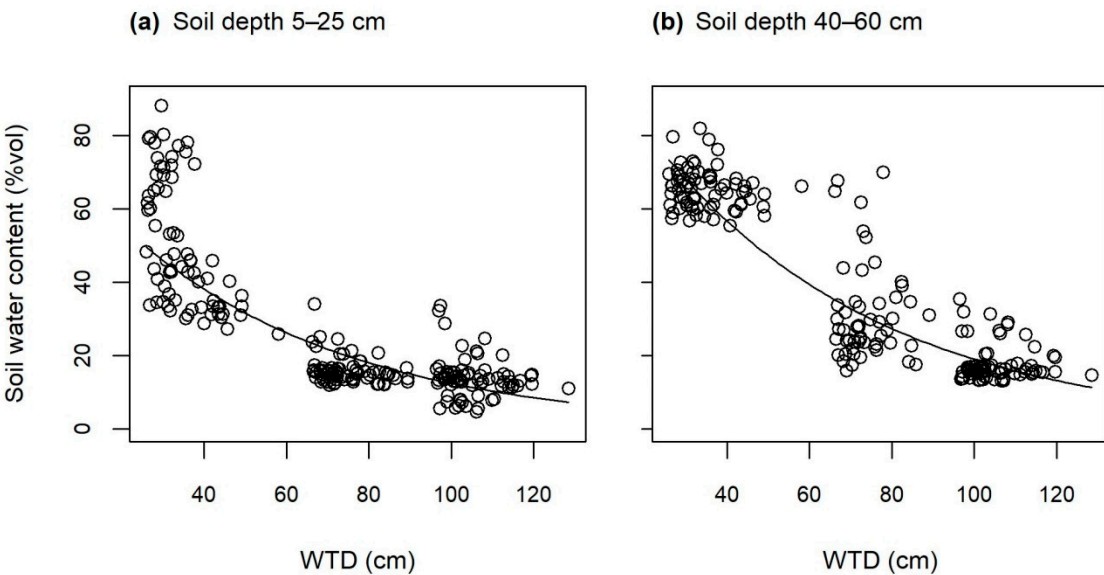

**Figure 3.** Soil water content in 5–25 cm (**a**) and 40–60 cm (**b**) soil depths depending on water table distance (WTD). Data were collected weekly in the second growing season from May to September 2019 at three positions in each tank (Figure 2). Lines are fitted by mixed effect LM: random effect = tank number (**a**): $\ln(y) = 4.39 - 0.02$ x, $p < 0.001$, $n = 216$; (**b**): $\ln(y) = 4.77 - 0.02$ x, $p < 0.001$, $n = 197$.

### 3.2. Growth and Aboveground Biomass

WTD had a significant effect on living aboveground biomass per plant in both years, although this effect was much smaller in the first than in the second year (Figure 4). Biomass was the highest at a WTD of around 50 to 60 cm (very to slightly moist soil) and achieved $167 \pm 49$ g in the first and $1491 \pm 410$ g in the second year (means $\pm$ standard deviation, dry weight). The latter was more than three times as high as in the driest treatment where only 458 g (mean, dry weight) living aboveground biomass per plant was measured. Living aboveground biomass was significantly determined by plant height (Spearman's rho = 0.75, $p < 0.001$) and not by number of shoots (Spearman's rho = 0.14, $p = 0.209$), considering the second year. Therefore, plant height was similarly affected by WTD as the living aboveground biomass and was between 135 and 335 cm (Figure 5). Plant height also reached its maximum at a WTD of around 50 to 60 cm (very to slightly moist soil) with $299 \pm 18$ cm in mean. Under wetter and drier soil conditions, plant height decreased.

Usually cup plant does not develop shoots before the second year [6]. However, in our experiment some individuals (34 of 84) had already developed one or more shoots (mean $1.6 \pm 1.0$) in the first growing season. This mainly occurred under moist soil conditions. Indeed, there was a significant correlation between number of shoot-developing individuals and row number of tank (Spearman's rho = $-0.92$, $p = 0.003$). In the second growing season, each individual independent of WTD developed from 8 to 32 shoots per plant (mean $18 \pm 5$). There was no significant effect of WTD on shoot number per plant (LM, $p = 0.714$).

All plants of all treatments grew and survived the two years of investigation in the experiment. However, at the end of the second growing season, a high portion of dead biomass in the three dry treatments was evident. There was in mean 23% and up to 73% (maximum) dead biomass in contrast to 6% (mean) in the wet, moist, and fresh treatments (Figure 6). There was a significant effect of WTD on the percentage of dead biomass.

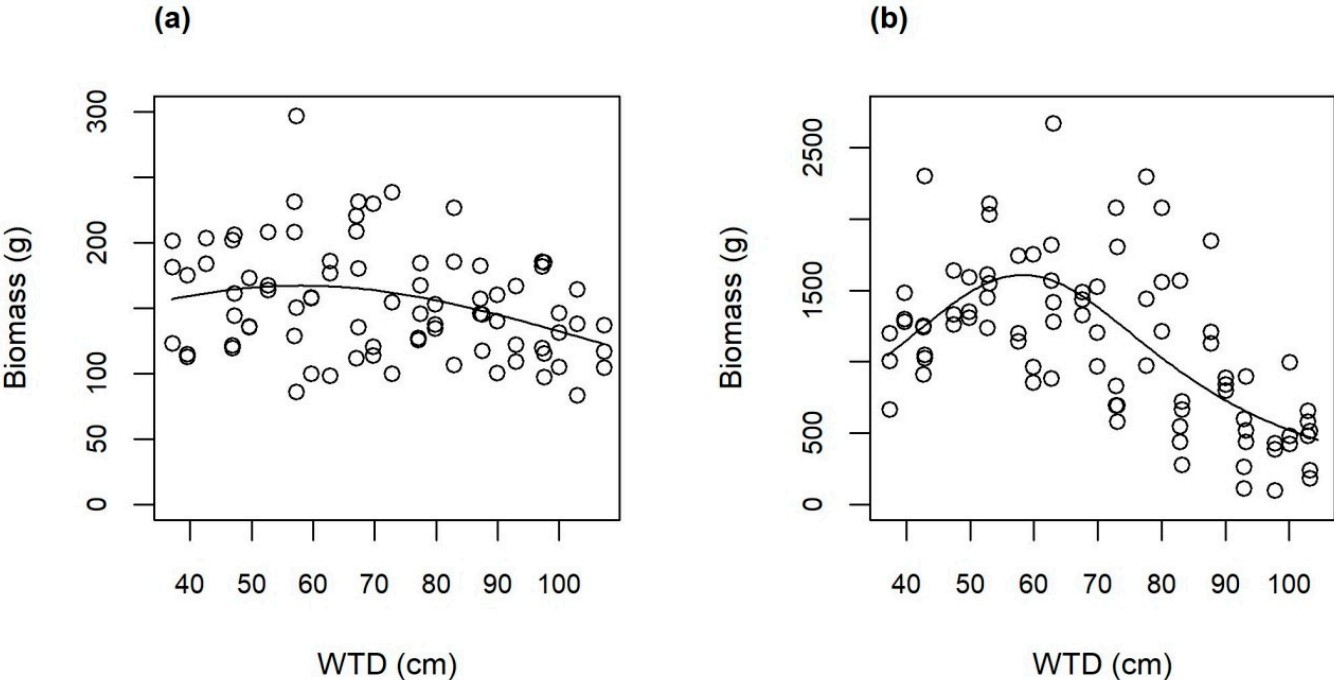

**Figure 4.** Living aboveground biomass (dry mass) per plant of cup plant, harvested (**a**): at the end of the first growing season (August 2018) and (**b**): at the end of the second growing season (September 2019) depending on water table distance (WTD). GLM: Gamma-distributed residuals, square function, (**a**): $p = 0.007$, $n = 82$; (**b**): $p < 0.001$, $n = 84$.

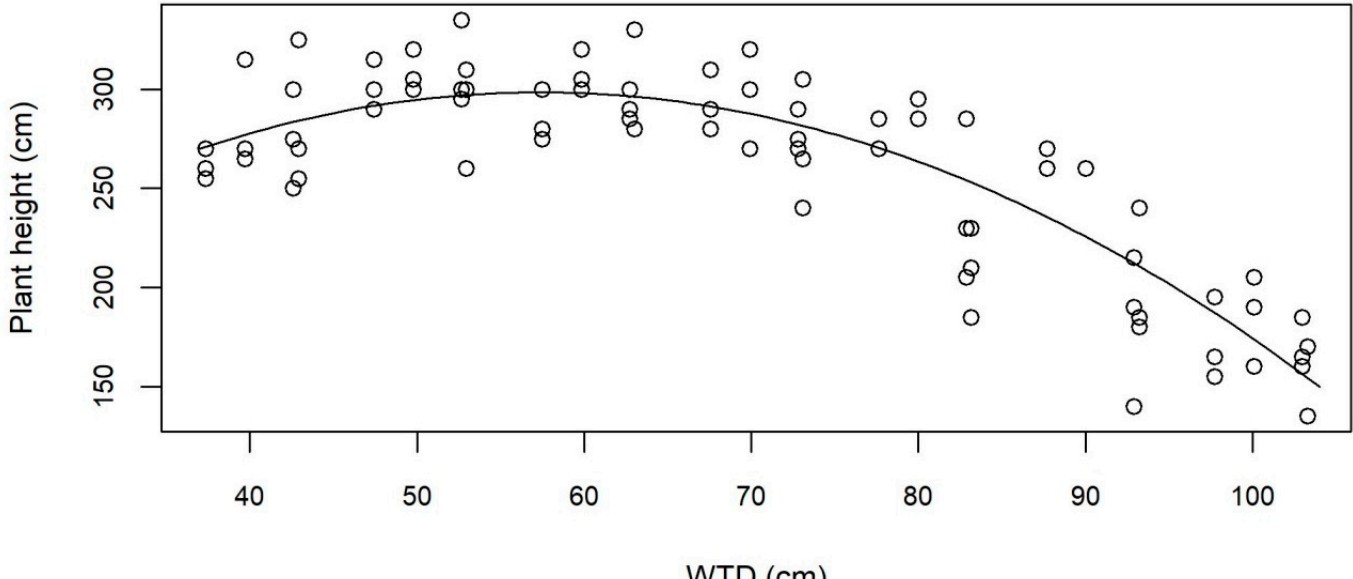

**Figure 5.** Plant height of cup plant at the end of the second growing season (September 2019) depending on water table distance (WTD). Mixed effect LM: random effect = tank number, square function, $n = 84$.

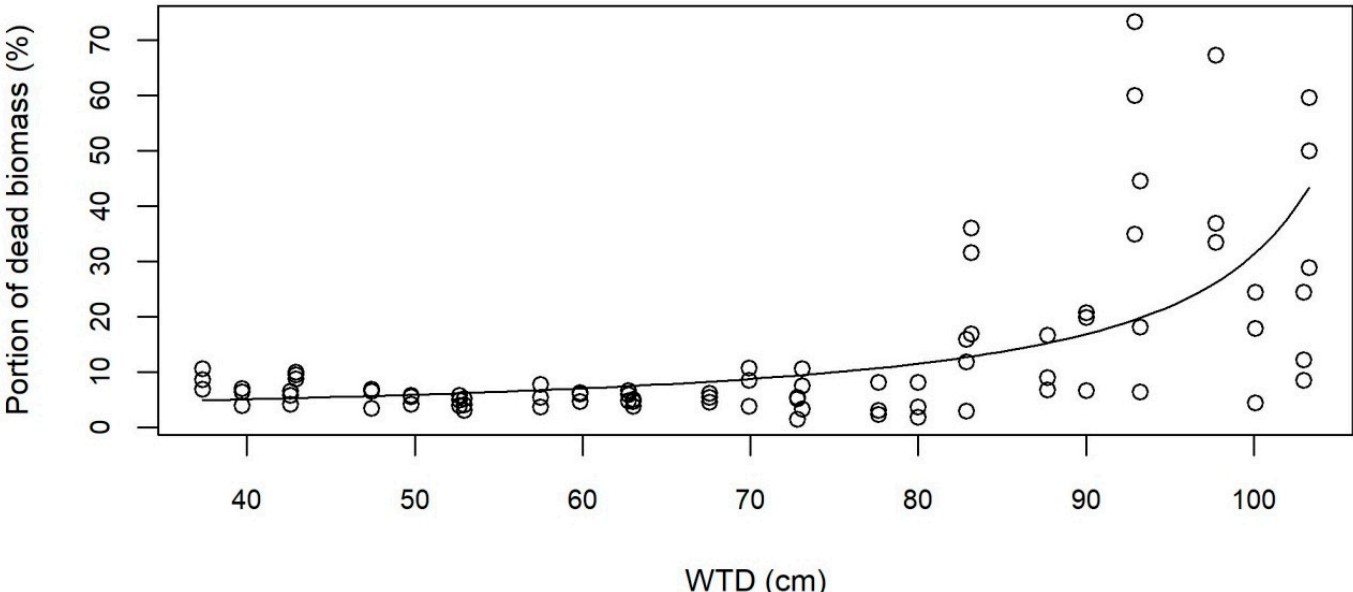

**Figure 6.** Portion of dead biomass of cup plant at the end of the second growing season (September 2019) depending on water table distance (WTD). GLM: Gamma- distributed residuals, $p < 0.001$, $n = 84$.

### 3.3. Reproductive Potential

There was a significant effect of WTD on the number of capitula at the end of the second growing season (Figure 7). The plants grown in very moist soil conditions (WTD around 50 cm) had the most capitula (mean $\pm$ standard deviation was 349 $\pm$ 156), whereas in wet soil conditions they developed slightly less (322 $\pm$ 143) and in rather dry soil conditions only a fifth (66 $\pm$ 115). In maximum one single plant developed 841 capitula (Figure 7). The number of fruits per capitulum was not affected by WTD (LM, $p = 0.734$) and was in mean 27 $\pm$ 4. The thousand grain weight was also not affected by WTD (mixed effect LM, $p = 0.115$) and was in mean 18.1 $\pm$ 3.9 g. Summing up, the plants grown in very moist to fresh soil conditions had a higher reproductive potential than those in dry or wet soil conditions because of a higher number of capitula. The number of capitula was significantly correlated with plant height (Spearman's rho = 0.64, $p < 0.001$) and not with number of shoots (Spearman's rho = $-0.02$, $p = 0.849$).

In September 2019, more than 90% of capitula of the plants grown in wet to fresh soil conditions had completed their flowering period and were already developing fruits (Figure 8). There was no significant difference between these four treatments regarding developmental stages of capitula (KruskalMC, $p > 0.05$). With increasing WTD the development slowed down. The drier the soil, the lower was the portion of fruiting capitula at the time of harvest and the higher was the portion of budding and flowering capitula. Regarding all treatments, there were significant correlations between row in tank and the portion of the three developmental stages of capitula (Spearman's rho for budding = 0.40, flowering = 0.68, fruiting = $-0.76$, $p$ always < 0.001). Thus, plants on drier soil conditions not only produced less capitula (Figure 7) but took also longer to develop them.

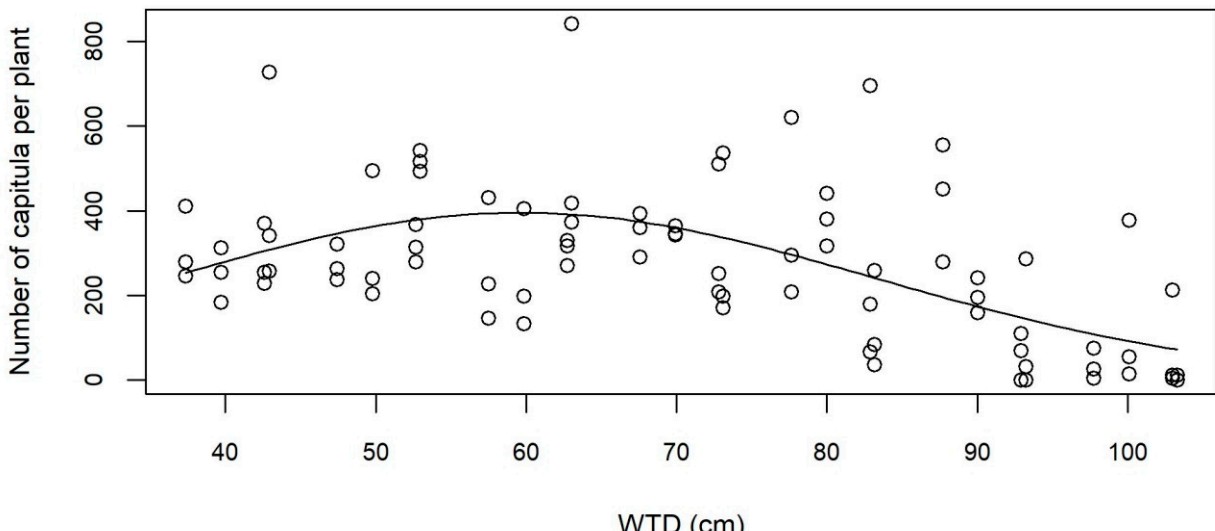

**Figure 7.** Number of capitula per plant of cup plant, regardless of their developmental stage, depending on water table distance (WTD). Data were collected at the end of the second growing season (September 2019). GLM: Poisson-distributed residuals, square function, $p < 0.001$, $n = 84$.

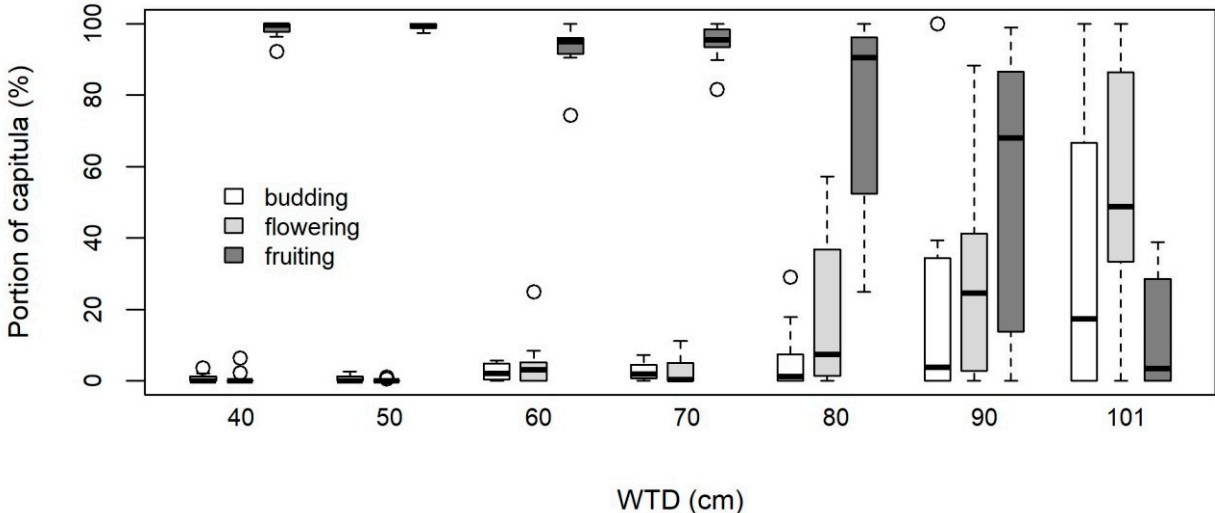

**Figure 8.** Portion of budding, flowering and fruiting capitula per plant of cup plant depending on water table distance (WTD, given as mean for each row of the four tanks). Data were collected at the end of the second growing season (September 2019). Note that the three different phenological stages of a treatment are always shown next to each other with a slight offset. $n = 12$ (in 90 cm WTD $n = 11$; in 101 cm WTD $n = 10$, because these three plants did not develop any capitulum at that time).

## 4. Discussion

### 4.1. Highest Yield on Moist Soil

Soil moisture conditions determined by water table distance (WTD) had a significant impact on growth and development of cup plant. Plant height and living aboveground biomass were the highest on moist soil with about 3 m and 1500 g dry weight per plant and decreased on wetter and especially on drier soil. Therefore, the plant height measured in our study reached the maximum published values for this species [19,29,30], indicating that optimal growth conditions were included in our study.

Several field studies and one pot experiment in Central Europe confirm our results that highest growth is achieved in periodically waterlogged or well-irrigated soil

conditions [5,31,32]. In general, highest yields of cup plant are described on soils with good soil moisture; hydromorphic soils are unsuitable [3,6]. Cup plant is able to reach deep water resources with its roots; therefore it is considered as certainly drought-tolerant [3]. Because of the limited soil depth in the present study, deep rooting was prevented and cup plant suffered considerable damage in the dry treatments. The number of shoots per plant was in mean 18 and not affected by soil moisture. This value is quite high compared to other studies, which indicated 3.5 to 6.6 shoots per plant in the second growing season [33,34]. Essential for this parameter are stand density and age of the plants [33,34]. The fact that cup plant develops shoots already in the first year, as shown in our study, has not been published so far. We assume that reasons are the sowing early in spring and the precultivation under optimal conditions in the greenhouse before planting into the experimental tanks.

A high yield of cup plant under moist soil conditions, as demonstrated in our study, is desirable from the farmers' point of view. However, from the perspective of invasion biology, this might carry the risk that spontaneously grown and established cup plants could also become such vigorous plants and might compete with native species. Studies are lacking but necessary to assess the competitiveness of cup plant and its possible risk of suppression of native species in case of spontaneous settlement. A species could be classified as invasive if its spread threatens biodiversity (Article 3 No. 2 EU-Regulation No. 1 143/2014).

### 4.2. High Reproduction and Rapid Development on Moist Soil

In the present study, cup plant produced the most capitula with about 350 on moist soil, and their development there was faster in comparison to drier soil conditions. Another study in Germany confirms our results, where the number of flowering capitula of cup plant was higher under irrigated than under rainfed conditions [35]. This study also agrees with ours concerning an independence of number of disc florets per capitulum in respect to watering. Although fruits of cup plant are developed from ray florets and not from disc florets [6], this agreement of results confirms that the composition of capitula is independent of soil moisture conditions. The number of fruits per capitulum was about 27 in the present study and therefore in the upper range or even above the values of other studies (10–20 or 20–30) [3,6]. Thousand grain weight varies widely in the literature (14 to 21.5 g [3] and up to 23 g [6]). Our values are with a mean of about 18.1 g rather in the middle.

The rapid development of fruits on moist soils leads to a high proportion of ripe fruits at harvesting time. Together with the high fruit production under these conditions, there is a higher risk of cup plant spreading from the fields—presupposed germination and saplings' establishment are likewise successful.

### 4.3. Consequences and Recommendations for Nature Conservation

Spontaneous occurrences of cup plant have already been documented in seven federal states of Germany and in other European countries as well [13–16]. From the view of nature conservation, the indication of colonized sites is important to assess the risk for protected or otherwise valuable ecosystems. In its native range in eastern North America, cup plant colonizes moist bottomlands, river valleys, and lakesides [3,6]. This is in line with our results and confirms a possible risk that cup plant could colonize moist habitats in Germany, too. So far, observations of spontaneous occurrences of cup plants in Germany have shown a broader range of habitats. In addition to ruderal places and woody structures, however, even moist ecosystems as perennial fields on river banks as well as bottomland woods are colonized [12,36,37]. This circumstance holds together with the high growth and reproductive potential on moist soils, as shown in our study a particular risk for nature conservation. Moist ecosystems—such as riparian fringes, alluvial forests, fens, and swamps—are valuable for nature conservation, because they are endangered according to the German red list of threatened habitats [38] and protected according to §30 BNatSchG.

Thus, an adequate distance of cup plant fields to moist ecosystems should be kept strictly to prevent their spontaneous colonization by cup plant. Dispersal distance of cup plant is 6 m in median but could be more than 10 m [12]. Therefore, we recommend for cup plant fields distances of several 10 m from valuable ecosystems to preclude fruit dispersal even under extreme wind events. However, dispersal vectors and distances of cup plant fruits are not investigated, and studies are urgently required to be able to give more precise recommendations for minimizing the risk of spreading. So far, it is also unknown whether cup plant fruits can be spread by watercourses and remain viable. As long as this is not examined, it is important to keep a sufficiently large distance to streams, even if they are strongly anthropogenic shaped and not valuable for nature conservation. In order to prevent fruit dispersal by agricultural machines, they should be cleaned before leaving the field and the crop should be covered during transport.

Additionally, the number and size of cup plant fields play a decisive role for the invasion potential because each newly cultivated field enhances the risk for further spontaneous spreading [12,39]. In Germany, more than 1000 ha are newly cultivated with cup plant in each of the recent years, while the older fields remain cultivated [40]. Consequently, further spreading of cup plant is to be expected and needs to be observed. The areas surrounding the cup plant fields and the roads from the fields to the farms should be continuously screened for spontaneous occurrences of cup plant to be able to combat this species where necessary.

## 5. Conclusions

In Central Europe, cup plant is a promising bioenergy crop that can achieve high yields, especially on moist soils. Wetter and drier soils are less suitable, but cup plant is able to survive on a wide range of soil moisture conditions.

However, a caution in respect to a possible invasiveness of cup plant is advised. The highest risk for spontaneous colonization by cup plant is—similar to the highest yield—supposed for ecosystems with moist soils, which are often valuable for nature conservation in Germany. To assess the actual invasive potential of cup plant, more studies about habitat requirements, competitiveness, and dispersal vectors of cup plant are urgently needed. If precautionary measures are observed, cup plant can take a place in the Central European agricultural landscape and make a valuable contribution to the conservation of biodiversity.

**Supplementary Materials:** The following tables contain the data on which calculations and figures of the present study are based. They are available online at https://www.mdpi.com/2077-0472/11/1/24/s1: Table S1: Weather conditions during the experiment; Table S2: Soil water content over time; Table S3: Soil water content of soil samples; Table S4: Growth of cup plant at the end of the first growing season (August 2018); Table S5: Shoot development of cup plant at the end of the first growing season (November 2018); Table S6: Growth of cup plant at the end of the second growing season (September 2019); Table S7: Explanation of column names of the Tables S1–S6.

**Author Contributions:** Conceptualization, M.L. and L.M.E.; methodology, L.M.E. and M.L.; validation, L.M.E. and M.L.; formal analysis, L.M.E.; investigation, K.K., M.K., A.J.F., and L.M.E.; data curation, L.M.E., M.K., K.K., and A.J.F.; writing—original draft preparation, L.M.E.; writing—review and editing, L.M.E., M.L., K.K., A.J.F., and M.K.; visualization, L.M.E.; supervision, L.M.E. and M.L.; project administration, L.M.E.; funding acquisition, L.M.E. and M.L. All authors have read and agreed to the published version of the manuscript.

**Funding:** This research was funded by the Oberfrankenstiftung and the District Government of Upper Franconia as well as the Studienstiftung des deutschen Volkes. The APC was funded by the German Research Foundation (DFG) and the University of Bayreuth in the funding program Open Access Publishing.

**Institutional Review Board Statement:** Not applicable.

**Informed Consent Statement:** Not applicable.

**Data Availability Statement:** The data presented in this study are available in the supplementary materials (Tables S1–S7).

**Acknowledgments:** We thank the Oberfrankenstiftung and the District Government of Upper Franconia for financial support as well as the Studienstiftung des deutschen Volkes for scholarship of the first author. Ralf Brodmann (Metzler & Brodmann Saaten GmbH) is thanked for free provision of cup plant seeds. We thank Andreas Schweiger (Plant Ecology, University of Hohenheim) for the support in developing the experimental design, Andreas Kolb (Soil Physics, University of Bayreuth) for the introduction to measurement technologies for soil water content and Oliver Archner as well as Stefan Holzheu (both Bayreuth Center of Ecology and Environmental Research BayCEER) for the installation of the measurement technology for water table. Special thanks are given to Frederik Werner, who supported us generously with plant cultivation and data collection as well as to the gardeners of the Ecological Botanical Gardens, who always assisted us with their expertise and manpower. Ursula Bundschuh (Soil Physics, University of Bayreuth) supported us with laboratory work of soil sampling. Micrometeorology group, University of Bayreuth and Bayreuth Center of Ecology and Environmental Research BayCEER is given thanks for providing meteorological data. Elisabeth Schaefer is thanked for English proof reading.

**Conflicts of Interest:** The authors declare no conflict of interest.

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
