# Peer review of "Possibly Invasive New Bioenergy Crop Silphium perfoliatum: Growth and Reproduction Are Promoted in Moist Soil"

_agriculture, doi:10.3390/agriculture11010024_

Round 1
Reviewer 1 Report
The authors investigated how the growth and reproductive fitness of Silphium perfoliatum depends on the distance from the groundwater table. They found that the plant preferes moist environments (although there was a slight growth reduction in very wet soil), and its growth is strongly reduced in drier soils. Interestingly the authors find that under opitmal conditions the plant produces shoots in its first year of growth which has to my knowledge not been described before.
While these results are clear, I am not sure that they alone can support the conclusion Silphium perfoliatum is a potentially invasive species capable of colonising various moist habitats. In agriculture it is generally necessary to apply herbicides during the first year of cultivation to prevent it from being overgrown with weeds, which indicates that Silphium seedlings cannot readily establish themselves in existing habitats. In line with this observation most reports of wild Silphium plants are in disturbed habitats, such as river banks and ruderal areas.
To answer the question if Silphium can invade moist habitats one would need to test a) if it can outcompete native species and b) if an established canopy of Silphium can prevent the normal ecological succession. Since the experimental setup used by the authors can be used to analyse the competitiveness of Silphium relative to native plants these questions could be answered, and I think this would greatly improve the value of the paper.
Reviewer 2 Report
The paper address an important issue - studying the "controversy" (or better: looking for a balance) between two "green" concerns - introducing the bioenergy production as an alternative to fossil fuels, and control of the plant invasions. Silphium perfoliatum provides an excellent example. The study is well organized, with very good experimental design and statistical treatment. The results are discussed in the light of similar studies, and in relation to the problems of control of plant invasions. I believe the paper will be of interest to the plant ecologists and conservation biologists, but also to people interested in the agricultural policy and sustainable agriculture.
I could provide the following two points that could be considered by the authors.
1. The only concern is about the recommendation for 20 m distance from "valuable ecosystems". It should be well-grounded and explained in some more details. I trust 100 % the authors' investigation on the maximum distance of distribution, but there could be some extreme events, like storms, or at least strong winds, which could change the situation.
2. Also, I believe the authors could mention an important recent study:
Bury M., Mozdzer E., Kitczak T., Siwek H., Włodarczyk M., 2020. Yields, calorific value and chemical properties of cup plant Silphium perfoliatum L. biomass, depending on the method of establishing the plantation. Agronomy (MDPI) 10(6), 851; https://doi.org/10.3390/agronomy10060851
Round 2
Reviewer 1 Report
The authors have addressed the concerns I had voiced in my first review. Although the paper does not fully answer the question of the invasiveness of the cup plant, it does certainly form a solid basis for further research.
